# Predicting Trajectories of Plate-Type Wind-Borne Debris in Turbulent Wind Flow with Uncertainties

Feng Wang [1], Peng Huang [2], Rongxin Zhao [1,*], Huayong Wu [1], Mengjin Sun [1], Zijie Zhou [1] and Yun Xing [1]

1 Shanghai Key Laboratory of Engineering Structure Safety, SRIBS, Shanghai 200032, China; wangfeng2@sribs.com (F.W.); wuhuayong@sribs.com (H.W.); sunmengjin@sribs.com (M.S.); zhouzijie@sribs.com (Z.Z.); xingyun@sribs.com (Y.X.)
2 State Key Laboratory of Disaster Reduction in Civil Engineering, Tongji University, Shanghai 200092, China; huangtju@tongji.edu.cn
* Correspondence: zhaorongxin@sribs.com

**Abstract:** Debris poses multifaceted risks and jeopardizes various aspects of the environment, human health, safety, and infrastructure. The debris trajectory in turbulent wind flow is more dispersed due to the inherent randomness of the turbulent winds. This paper investigates the three-dimensional trajectories of plate-type wind-borne debris in turbulent wind fields via the method of numerical simulation. A 3D probabilistic trajectory model of plate-type wind-borne debris is developed. The debris trajectories are numerically calculated by solving the governing equation of debris motion and by introducing turbulent wind flows based on the near-ground wind field measured in the wind tunnel to account for the probability characteristics of the debris trajectory. The dimensionless velocities and displacements of the debris trajectory show good agreement with the experimental data in wind tunnel tests, confirming the rationality of the probabilistic trajectory model. Based on the validated trajectory model, the probability characteristics of the debris impact position, impact velocity, and kinetic energy, debris angular displacement, and angular velocity are analyzed in detail under five different wind attack angles. The proposed probabilistic model of plate-type debris in turbulent wind flow provides an accurate and effective method for predicting debris trajectory in three-dimensional space.

**Keywords:** numerical simulation; plate-type debris; turbulent flow; three-dimensional trajectory; low-rise building





## 1. Introduction

Wind-borne debris is one of the most dangerous factors during extreme wind storms, as it can destruct habitat as well as strike human beings [1–5]. Thus, it is of great significance to understand the mechanism of debris release and trajectory characteristic in order to decrease casualties and economic losses. The damage caused by wind-borne debris is complex and changeable with different debris conditions, wind conditions, aerodynamic characteristics of debris, and strength of the impacted structure [6–9]. Field measurement and wind tunnel testing of wind-borne debris damage is difficult and costly to implement. In this situation, numerical methods become more and more important in solving this problem.

The trajectories of plate-type debris were studied in both wind tunnels and numerical simulations in [10–12]. Good agreement between the calculated and measured trajectories was obtained when lift forces proportional to the rate of rotation (the "Magnus" effect) were incorporated into the calculations. Lin et al. [10–13] verified the impact of the Tachikawa number on debris flight trajectories via numerous wind tunnel tests. Baker [14] studied the two-dimensional motion of both compact and plate-type debris via numerical simulation and proposed an alternative non-dimensional scheme to that of Tachikawa. For compact debris, the analytical results of Baker agreed well with the flight trajectories predicted

by [15]; However, they were 12% smaller than the wind tunnel and numerical results in [16], in which the vertical aerodynamic drag was ignored. For plate-type debris, the analytical results of [14] agreed well with wind tunnel test results of [17] at large angles of initial inclination, whereas the approaches have some divergence at small initial inclinations.

Richards et al. [18–20] utilized numerical methods to solve the classical Euler dynamics equation and developed a six degree of freedom (DoF) deterministic debris flight model. The forces on the sheet-debris in this model were derived from wind tunnel testing. Similar to the research of [18,19], Noda and Nagao [20] analyzed the force coefficient of a plate with various wind directions. They investigated the effects of the Tachikawa number and plate aspect ratio on the debris trajectory.

With the development of Computational Fluid Dynamics (CFD), researchers have tried to simulate the flow characteristics of blunt bodies on computers. Martinez-Vazquez et al. [21] utilized CFD and analytical methods to study the trajectory of plate-type debris. Kakimpa et al. [22,23] innovatively combined CFD with rigid body dynamics (RBD) and developed the CFD-RBD model to examine the three-dimensional trajectories of plate-type debris. The CFD-RBD model has a high degree of accuracy, allowing it to simulate the three-dimensional trajectories of debris. Moghim and Caracoglia [24] developed a mean velocity field varying with elevation and turbulence to determine the debris in a complex flow field. In addition, Huang et al. [25] proposed a three-dimensional fluctuating wind flow to investigate the probabilistic trajectories of plate-type wind-borne debris, Sabharwal and Guo [26] proposed a Stereophotogrammetry method to track the 6-DOF flight trajectory of wind-borne debris.

Visscher et al. [27–29] carried out the wind tunnel test at the University of Western Ontario to study the flight of aeroelastically-scaled plates mounted on the roof of a low-rise building model in a scaled atmospheric boundary layer flow, then analyzed the effects of the initial wind direction angle and initial flight condition on the trajectories of sheet-debris. Their results showed that all of the possible modes of flight observed by Tachikawa can occur under the same nominal initial conditions when the plate is mounted on a building surface and fails under turbulent wind loads. Moghim et al. [30] carried out a wind tunnel test in Northeastern University's small-scale wind tunnel under both smooth flow and grid-generated turbulent flow. The motion of spheres and cubes, simulating compact debris objects, was investigated on a vertical plane in two dimensions. The experimental results were shown to be consistent with the numerical simulations. The authors confirmed that both the mean flow speed and the turbulence features can have non-negligible effects on the trajectory of compact objects.

Obviously, the experimental method in the study of debris trajectory is more accurate and effective; however, it is limited and costly at the same time [28,30]. In consequence, the numerical method is frequently used in the simulation of debris trajectory. The existing research on debris trajectories obtained by the numerical method is either in two dimensions, [14], in a uniform flow field [19], or incurs a heavy computational load [22,23]. Therefore, there is a need to find a balance between the accuracy and efficiency of the computation.

In this study, a 3D probabilistic trajectory model of plate-type wind-borne debris is developed. The trajectory of the debris is obtained by numerical integration of the governing equations of motion of the debris, while the wind speed histories measured in the wind tunnel are imported into the governing equations to account for the turbulence of the wind field. The accuracy of the model is verified by comparing the dimensionless velocity and displacement of debris trajectory with the results of other experiments, and the probability characteristics of the trajectories of plate-type wind-borne debris are investigated based on the validated trajectory model. The debris flight equations and initial conditions are presented in Section 2. The simplified turbulence flow obtained by measurement in the wind tunnel and the rationality of the debris trajectory simulation results are discussed in Section 3. Section 4 investigates the probabilistic flight trajectories of debris under five different wind attack angles by analyzing the debris impact position, impact velocity, kinetic

energy, angular displacement, and angular velocity in three-dimensional space. Finally, general conclusions are drawn in Section 5.

## 2. Debris Flight Trajectory Model Establishment

The core of the numerical simulation of debris flight trajectory is to solve the Classical Newton–Euler equations of the rigid body using the Runge–Kutta method. The classical Newton–Euler equations of the rigid body are:

$$m\frac{d\mathbf{V}}{dt} = \mathbf{F}_g \tag{1}$$

$$\mathbf{I}_p\frac{d\boldsymbol{\omega}_p}{dt} = \mathbf{M}_P - \boldsymbol{\omega}_p \times \mathbf{I}_p\boldsymbol{\omega}_p \tag{2}$$

where $m$ represents the mass of the body, $t$ is the time, $\mathbf{V}$ is the velocity vector, $\mathbf{F}$ is the force vector, $\mathbf{I}$ is the angular momentum vector, $\mathbf{M}$ is the moment vector, and $\boldsymbol{\omega}$ is the angular velocity vector. The subscripts $g$ and $p$ are used to denote a vector expressed in a global inertial reference frame ($Xg$, $Yg$, $Zg$) and body-fixed coordinate ($Xp$, $Yp$, $Zp$), respectively. As shown in Figure 1, $X$, $Y$, and $Z$ represent the longitudinal, vertical, and lateral directions, respectively. $\mathbf{U}p$ is the relative speed of the debris to the wind in a body-fixed coordinate system and $\mathbf{V}$ is the debris flight speed in the global inertial reference frame.

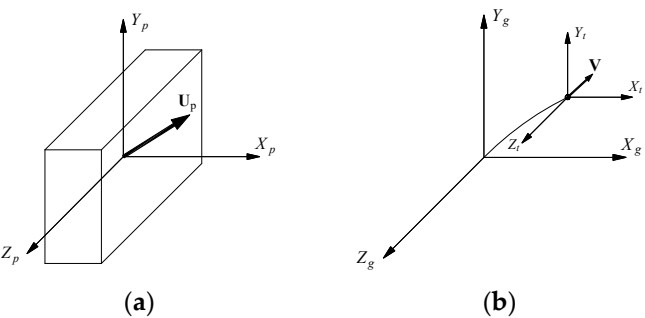

|  |  |
|:---:|:---:|
| (**a**) | (**b**) |

**Figure 1.** (**a**) Body-fixed coordinate system ($Xp$, $Yp$, $Zp$) and (**b**) global inertial reference frame ($Xg$, $Yg$, $Zg$) and transnational coordinate ($Xt$, $Yt$, $Zt$).

Moving from a velocity vector in the global inertial reference frame to a body-fixed coordinate system first requires a translation and then a rotation of the transnational coordinates. Extensive research describes this rotation using Euler angles ($\varphi$, $\theta$, $\psi$). However, singularities arise when the parameters of the Euler angles are utilized in numerical integration, and this can lead to gimbal lock and reduced computational efficiency, as explained in [25,31–34]. Fu et al. [31] proposed a rotational quaternions method to transform the debris orientation from a global inertial reference frame to a body-fixed reference frame, as follows:

$$\mathbf{q} = \begin{bmatrix} q_0 \\ q_1 \\ q_2 \\ q_3 \end{bmatrix} = \begin{bmatrix} \cos(\alpha/2) \\ p_1\sin(\alpha/2) \\ p_2\sin(\alpha/2) \\ p_3\sin(\alpha/2) \end{bmatrix} \tag{3}$$

The normal force coefficients ($\mathbf{C}_F$) on the plate debris are derived from the research of [18], as shown in Figure 2, and the forces ($\mathbf{F}_P$) on the plate are calculated as follows:

$$\mathbf{F}_P = \mathbf{C}_F(\varepsilon,\gamma,G)\frac{1}{2}\rho_a|\mathbf{U}_p|^2 A_r = \begin{pmatrix} F_{PX} \\ F_{PY} \\ F_{PZ} \end{pmatrix} = \frac{1}{2}\rho_a|\mathbf{U}_p|^2 \begin{pmatrix} C_{FX}(\varepsilon,\gamma,G)l_Yl_Z \\ C_{FY}(\varepsilon,\gamma,G)l_Xl_Z \\ C_{FZ}(\varepsilon,\gamma,G)l_Xl_Y \end{pmatrix} \tag{4}$$

where $\mathbf{C}_F(\varepsilon, \gamma, G)$ is the force coefficient at a wind attack angle of $\varepsilon$ and a tilt angle of $\gamma$ with a debris aspect ratio of $G$, while $\rho_a$ is the air density, $Ar$ is the reference area of the debris, and $l_X$, $l_Y$, and $l_Z$ are the size of the debris in the $X$, $Y$, and $Z$ directions, respectively.

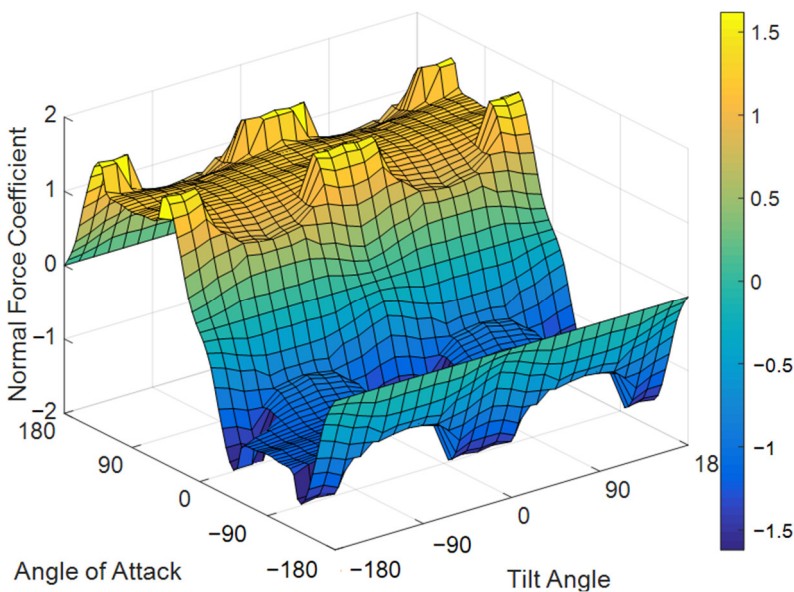

**Figure 2.** Normal force coefficients for plates with side length ratios equal to 2 [18].

The hysteresis effects resulting from dynamic stall and apparent camber are considered as follows:

$$\Delta C_N = \begin{cases} \frac{2\pi}{1+2/AR} \min(\frac{d\varepsilon}{dt} \frac{c\cos(\varepsilon)}{2|\mathbf{U}_P|}, 0.4), & \frac{d\varepsilon}{dt} \cdot \cos(\varepsilon) \geq 0 \\ \frac{2\pi}{1+2/AR} \max(\frac{d\varepsilon}{dt} \frac{c\cos(\varepsilon)}{2|\mathbf{U}_P|}, -0.4), & \frac{d\varepsilon}{dt} \cdot \cos(\varepsilon) < 0 \end{cases} \tag{5}$$

where $\Delta C_N$ is the change in the normal force and $AR = b^2/A, b = l_Z \times |\cos(\gamma)| + l_Y \times |\sin(\gamma)|$.

The total moments ($\mathbf{M}_P$) applied to the plate include the external applied moment ($\mathbf{M}_E$) and damping moment ($\mathbf{M}_D$).

$$\mathbf{M}_p = \mathbf{M}_E + \mathbf{M}_D \tag{6}$$

$$\mathbf{M}_E = \mathbf{F}_P \cdot d_{cop} \tag{7}$$

$$\mathbf{M}_D = \mathbf{C}_{DM} \frac{1}{2} \rho (|\mathbf{U}_P| + |\boldsymbol{\omega}_p| l/2) A_r l^2 \boldsymbol{\omega}_p \tag{8}$$

The center of pressure ($d_{cop}$) and drag coefficient ($\mathbf{C}_{DM}$) result from the wind tunnel tests of [18] performed at the University of Auckland.

The force on debris $\mathbf{F}_p$ should be transferred from the body-fixed coordinate into the translational coordinate; then, the translational acceleration of the debris mass center is derived as follows.

$$m \frac{d\mathbf{u}_g}{dt} = \mathbf{F}_t - mg\mathbf{j} \tag{9}$$

In this paper, $\mathbf{U}$ is measured by the wind tunnel; it stands for the turbulent wind field the plate debris is immersed in, and is explained in Section 3.2. The flight trajectory of the debris is obtained by introducing $\mathbf{U}$ into Equation (9) and solving the coupled governing Equations (2) and (9) through the fourth-order Runge–Kutta integral method, and the integral is stooped when the debris impacts the ground.

## 3. Wind Speed Experiments for Debris Trajectory Predictions

### 3.1. Wind Tunnel Test of Plate-Type Debris Environments

It has previously been observed that the turbulence of the wind flow has significant effects on the debris flight [15,24,27]. Existing numerical analyses on debris flight trajectory regard the wind flow either as a uniform wind flow or as a turbulent flow in two dimensions, and as such cannot represent the complex wind flow in a realistic situation. To investigate the three-dimensional trajectories of plate-type wind-borne debris in a turbulent flow around a low-rise building, the trajectories of the debris were simulated by a numerical method and a wind tunnel test was carried out at the University of Birmingham to validate the accuracy of the trajectories. Owing to spatial constraints, this paper only discusses the results of the numerical simulation.

In what follows, a low-rise building model with dimensions (length, height, and width) of $0.2 \times 0.213 \times 0.2$ m and plate-type debris with dimensions of $0.02 \times 0.0008 \times 0.04$ m is investigated. The building model has a roof pitch of 1:3 and the height to the eaves is 0.18 m. The debris are made of balsa wood with a weight of 0.2 g and a density of 312.5 kg/m$^3$. Figure 3 shows the definition of the wind direction and initial (starting) position of the debris. These dimensions were chosen because the building represents a 1:30 scale model, which stands for a typical low-rise building and is subsequently used in a series of wind tunnel experiments. At a scale ratio of 1:30, the equivalent full-scale dimensions of the low-rise building are $6 \times 6.39 \times 6$ m and the plate-type debris are $0.6 \times 0.024 \times 1.2$ m, and the results of debris trajectory shown below are in full-scale.

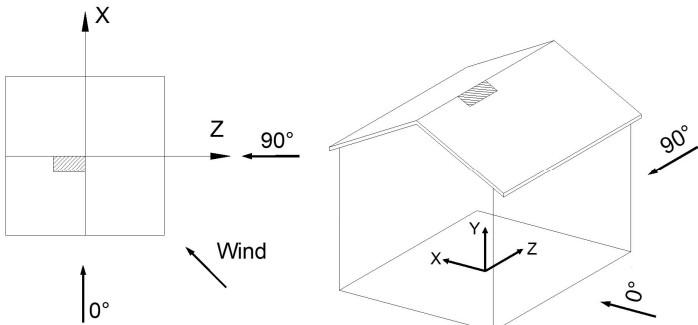

**Figure 3.** Illustration showing the model building and definition of wind directions and initial debris position.

### 3.2. Experimental Measured Turbulent Wind Flow

The debris trajectory experiment was carried out under a wind tunnel simulated atmospheric boundary layer flow field; the velocity profile and turbulence intensity profile of the approaching wind flow are shown in Figure 4. The mean wind speed increases with the measuring height, and increases from 3.73 m/s to 7.61 m/s and 8.6 m/s when the measuring height is 0.5 m and 1.5 m, respectively. By contrast, the turbulence intensity decreases with the measuring height; it has a value of about 24% when the height is below 0.1 m. It quickly decreases to 7.61% at the height of 0.5 m and slowly decreases to 2.12% at the height of 1.5 m. The roughness exponent of the wind speed velocity is 0.20 and the mean values of the longitudinal, lateral, and vertical turbulence integral scale of the wind field are 0.92, 1.09, and 1.27 m, respectively.

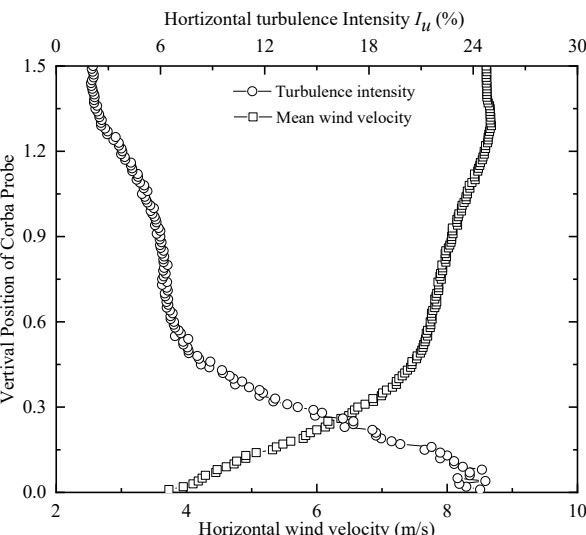

**Figure 4.** Velocity profile and turbulence intensity profile of the approaching wind flow.

The wind speed histories of the three components were measured by a hot cobra with a sampling frequency of 250 Hz at the release position of the debris; this point was chosen as it best reflects the flow field in which the debris is immersed. The mean wind speed, standard deviation of the fluctuating wind speed, and turbulence intensity of the wind speed histories are defined as follows:

$$W_i = \frac{1}{N} \sum_{j=1}^{N} w_i(t), \, i = x, y, \text{ and } z \tag{10}$$

$$\sigma_i = \sqrt{\frac{\sum_{j=1}^{N}(W_i - w_i(t))}{N}} \tag{11}$$

$$I_i = \frac{\sigma_i}{W_t} \tag{12}$$

where $N$ is the number of samples, $w_i(t)$ is the wind speed history of the three components measured by the hot cobra, $\sigma_i$ is the standard deviation of the fluctuating wind speed, and $W_t = \sqrt{W_x^2 + W_y^2 + W_z^2}$ is the total mean wind speed.

As the flight time of the debris in the above-mentioned wind tunnel test was less than one second, the changing flow field is expressed in the short period of the wind speed time history. In this way, the computation efficiency is improved and we only need to guarantee the computational accuracy of the method. Figure 5 shows example time histories of the longitudinal ($W_x$), vertical ($W_y$), and lateral ($W_z$) wind speed at the initial debris release position in the wind tunnel with a length of three seconds. The 3 s mean wind speeds in the longitudinal, vertical, and lateral directions are 4.34, −0.11, and 0.14 m/s, respectively, for a total wind speed of 23.8 m/s at full scale. Figure 6 displays the spectrum of the fluctuating wind velocity of all three components at the debris release point in comparison with that of the Von Karman normalized wind speed spectrum, which is shown in Equations (13) and (14). It can be seen that the spectra of the longitudinal, lateral, and vertical fluctuating wind velocities all correspond well with the Von Karman spectra and are mostly according to realistic wind spectra. As the wind speed history measured in wind speed expresses the flow field in which the debris are immersed, the trajectories of the

plate-type wind-borne debris can then be obtained by inputting the wind speed histories into Equations (1), (2), and (9) and performing numerical integration.

$$\text{Longitudinal direction}: \quad \frac{nS_x(n)}{\sigma_x^2} = \frac{4f}{(1+70.8f^2)^{5/6}} \tag{13}$$

$$\text{Vertical, and lateral direction}: \quad \frac{nS_i(n)}{\sigma_i^2} = \frac{4f(1+755.2f^2)}{(1+283.2f^2)^{11/6}}, \quad i=y,z \tag{14}$$

where $S(n)$ is the power spectral density function of the fluctuating wind speed and $f$ is the frequency.

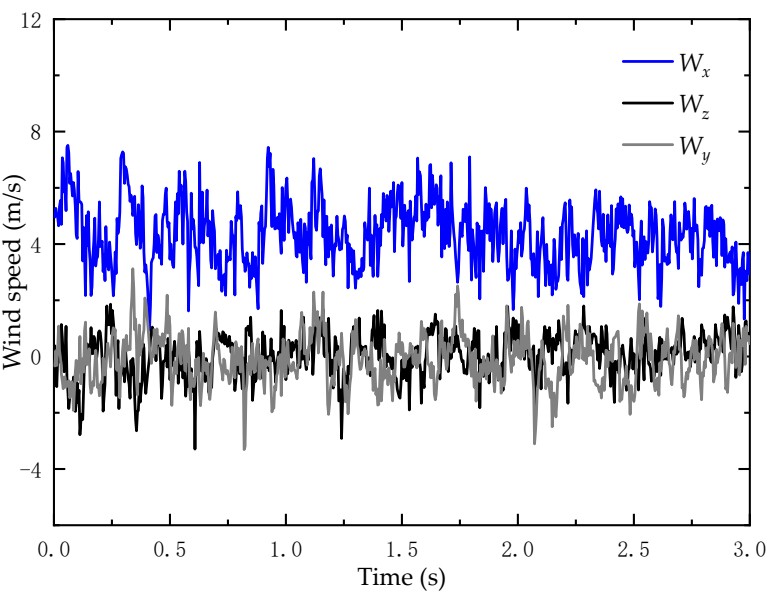

**Figure 5.** An example of wind speed history measured at the initial debris release position in the wind tunnel.

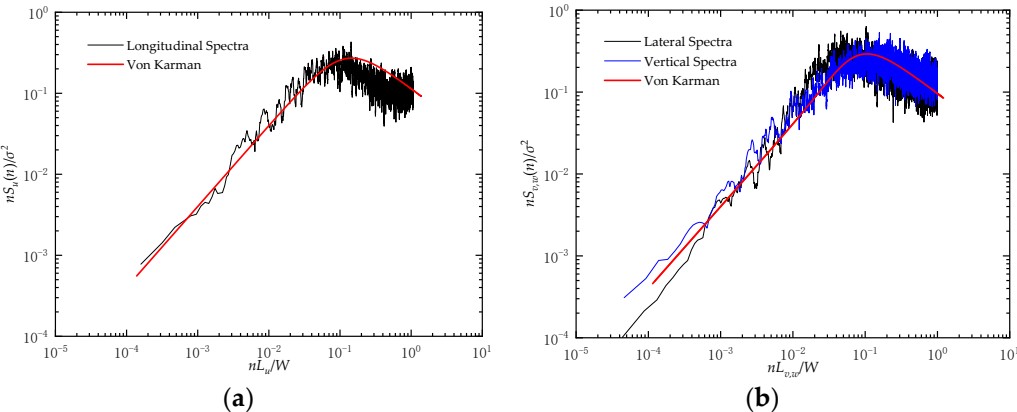

**Figure 6.** (**a**) Longitudinal and (**b**) lateral and vertical spectra of the fluctuating wind velocity at the debris release point.

### 3.3. Rationality of the Trajectory Simulation

The numerical simulations presented in this paper are based on the wind speed histories and force coefficients measured in the wind tunnel. This approach maximally reproduces the probabilistic trajectories of plate-type debris in wind tunnel tests, though it may be different from the debris in full-scale field measurement [35]. To validate the rationality of the numerical results in this paper, the dimensionless flight velocity of the debris is compared with the results derived from both numerical simulations and wind

tunnel testing by other researchers. For convenience of comparison, the Tachikawa number and dimensionless flight velocity, displacement, and flight time are defined according to the research of [17], as follows.

$$K = \frac{\rho W^2}{2mhg} \tag{15}$$

$$\overline{u} = \frac{V}{W} \tag{16}$$

$$\overline{x} = \frac{gx}{W^2} \tag{17}$$

$$\overline{y} = \frac{gy}{W^2} \tag{18}$$

$$\overline{t} = \frac{gt}{W} \tag{19}$$

Lin et al. [11] carried out a series of wind tunnel tests and proposed the fitting formula of plate-type debris to show the relationship between dimensionless velocities, displacement, and flight time as Equations (20) and (21).

$$\overline{X} = 1 - e^{-\sqrt{1.8K\overline{x}}} \tag{20}$$

$$K\overline{x} \approx 0.456(K\overline{t})^2 - 0.148(K\overline{t})^3 + 0.024(K\overline{t})^4 - 0.0014(K\overline{t})^5 \tag{21}$$

Baker [14] presented the approximate expression for plate-type debris based on his analysis results:

$$\overline{X}_1 = 1 - e^{(-1.2\overline{x}_1^{0.5})} \tag{22}$$

where $\overline{X}_1$ and $\overline{x}_1$ are the dimensionless debris flight velocity and longitudinal displacement defined by [14]. Transferring Baker's equation with the dimensionless parameter defined by [17], the equation from [14] is equal to

$$\overline{X} = 1 - e^{(-2.7\overline{x}^{0.5})} \tag{23}$$

Figure 7a shows the nine cases of the simulated results of the relationship between dimensionless velocity and debris displacement along with a comparison with the wind tunnel test of [11] and numerical analysis of [14]. As would be expected, the results presented in this paper correspond well with both of the other two results. In addition, the analysis results from [14] are smaller than the wind tunnel test results of [13]. This is probably because the analysis results from [14] are based on uniform wind fields and the wind tunnel results include the turbulence flow field. These results may indicate that the numerical results of debris flight trajectories can predict the mean value of debris flight distances and velocities. Therefore, it is unsafe not to consider the turbulence wind flow on debris flight, as most debris damage events are caused by extreme situations.

Figure 7b presents the dimensionless flight distance varying with flight time and a comparison with the fitting formula from [11]. It can be seen that the simulated results in this paper coincide reasonably with the wind tunnel results. These comparisons guarantee the rationality of the simulation method presented in this paper.

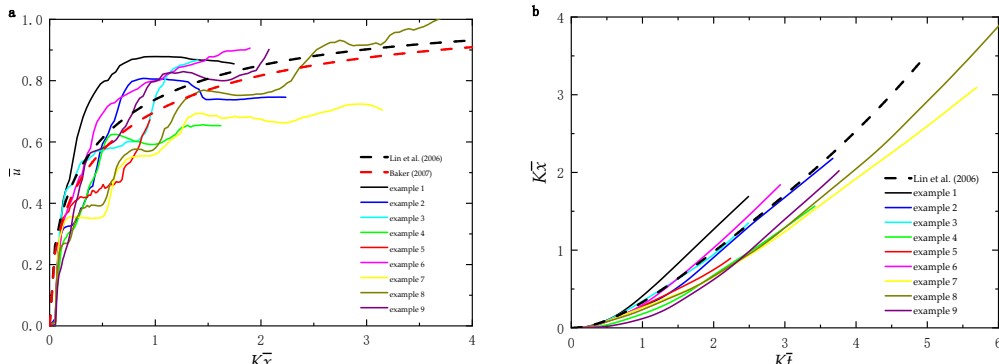

**Figure 7.** Nine examples of (**a**) dimensionless displacement vs. dimensionless velocity and (**b**) dimensionless flight time vs. dimensionless displacement (compare with Lin et al., 2006) [11,14].

## 4. Characteristics of Debris Flight

As the rationality of the method of implying experimental wind speed in the calculation of the debris flight trajectory has been discussed in Section 3, this section focuses on the characteristics of debris trajectory with the same initial condition as in Section 3. To investigate the probabilistic of debris trajectory, one hundred wind speed samples of 1 s each measured in a wind tunnel were selected; the mean wind speeds, turbulence intensities, and turbulence integral scales of the wind speed history are shown in Figure 8. From the figure, it can be seen that the difference in mean wind speed, turbulence intensity, and longitudinal and vertical turbulence integral scales between the 100 cases are small and that the variation of the lateral integral scale is larger than that in the longitudinal and vertical directions. In addition, the probability density distribution of horizontal mean wind speed, turbulence intensity, and turbulence integral scale of the one hundred 1 s wind speed time histories fit well with the Generalized Extreme Value Distribution (Gevfit).

### 4.1. Impact Position and Impact Velocity

Figure 9 presents the 100 debris trajectories in wind attack angles of 0°, 15°, 30°, 45°, and 60°. These results indicate that the wind attack angle has a large effect on the debris flight velocities and trajectories. At a 0° wind attack angle, with the debris landing in a relatively narrow lateral displacement, the debris impact velocity increases with longitudinal displacement and many pieces of debris impact the ground with a dimensionless velocity larger than 1. The debris displacements and flight speeds are smaller in wind directions of 15° and 30° compared with that in the 0° direction. When the wind attack angle increases to 45° and 60°, the debris has a larger vertical and lateral displacement, and the debris horizontal flight velocity of many trajectories first increases then decreases with the longitudinal displacement.

Figure 10 displays how the positions at which the debris impacts the floor vary with the lateral integral scale; the color bar stands for different ranges of the lateral integral scale. The landing positions of the debris are more concentrated in smaller wind attack angles, and increase with the wind attack angles. Figure 6 shows that the lateral integral scale is the most apparent difference between the 100 wind speed time histories, whereas the lateral integral scale makes no major difference in the landing positions of the debris, as shown in Figure 8.

The damage when the debris strikes a building is assumed to be proportional to the kinetic energy of the debris. The kinetic energy of the debris is defined as

$$E = \frac{1}{2}mV_t^2 \tag{24}$$

where $m$ is the mass of the debris and $V_t$ is its total velocity.

Thus, the dimensionless kinetic energy can be expressed as follows.

$$\overline{E} = \frac{E}{0.5mW_t^2} = \left(\frac{V_t}{W_t}\right)^2 \tag{25}$$

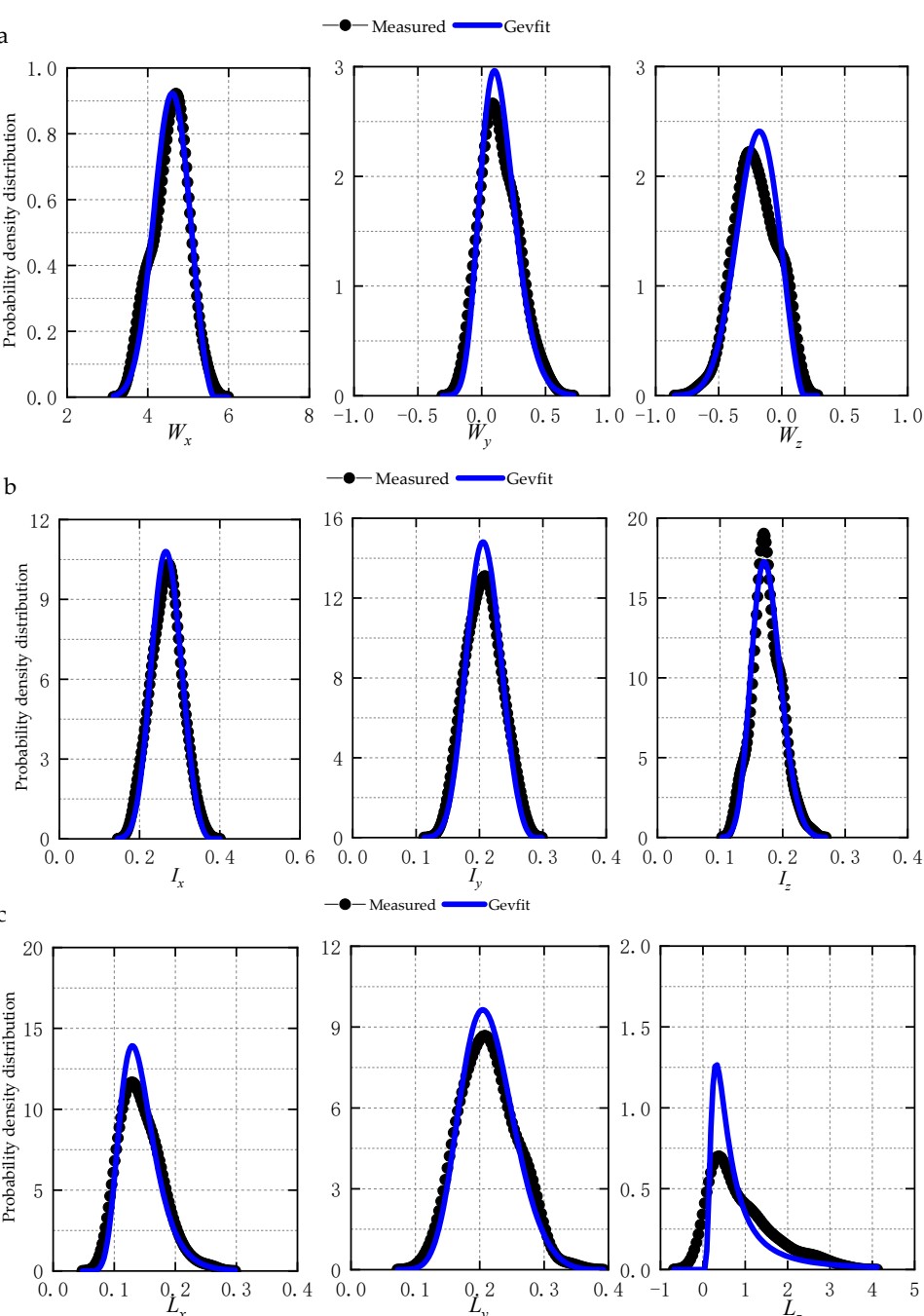

**Figure 8.** Probability density distribution of (**a**) mean wind speed, (**b**) turbulence intensity, and (**c**) turbulence integral scale of the one hundred 1 s wind speed time histories.

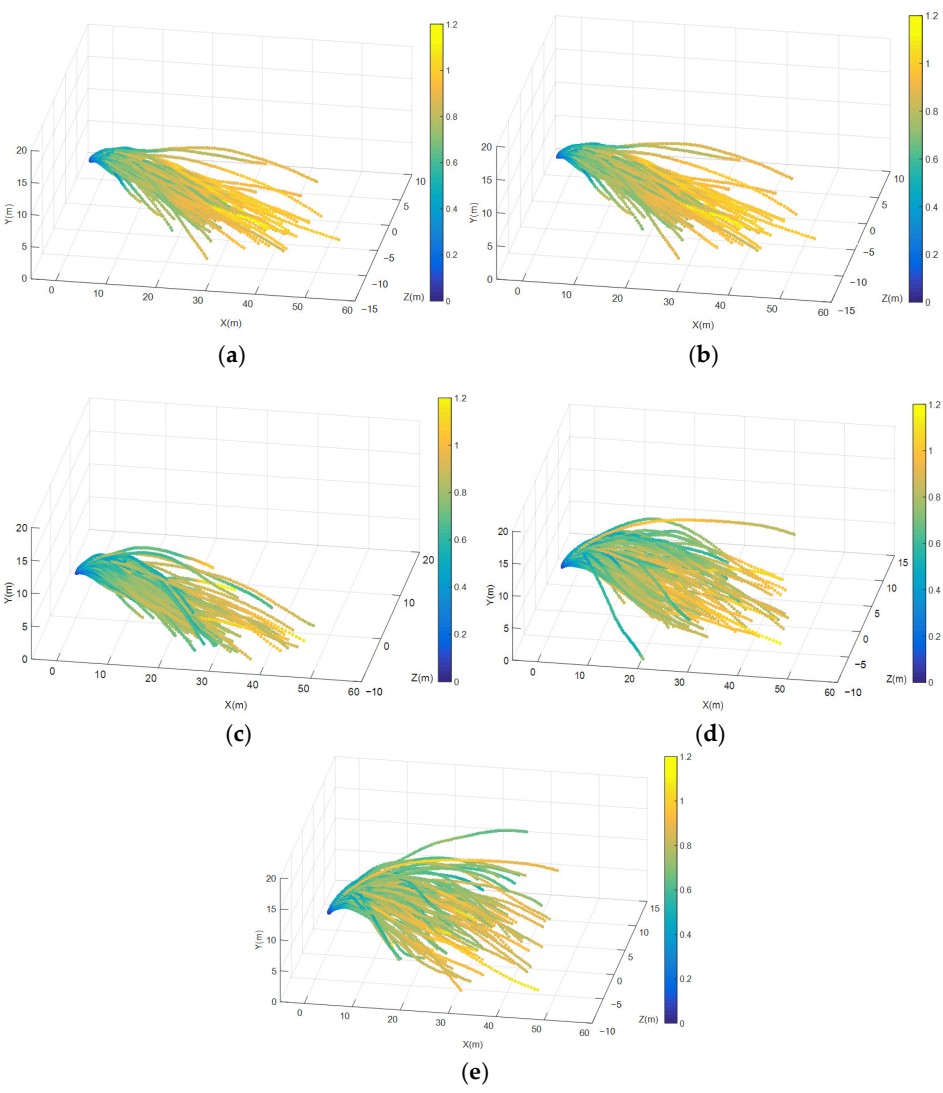

**Figure 9.** One hundred examples of positions of debris center and the dimensional horizontal flight velocity of the debris at wind attack angles of (**a**) 0°, (**b**) 15°, (**c**) 30°, (**d**) 45°, and (**e**) 60°.

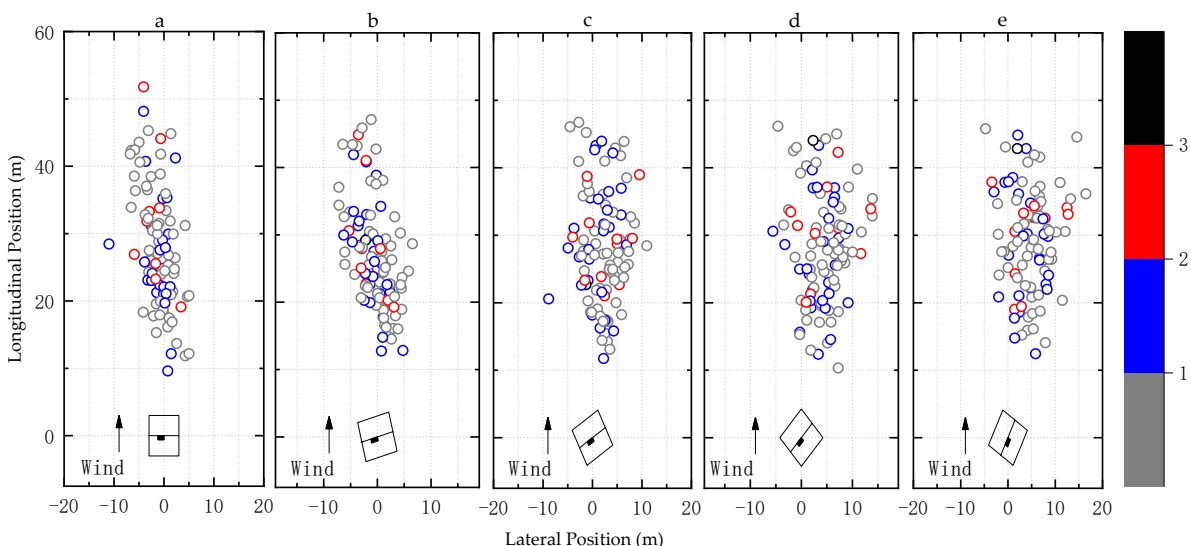

**Figure 10.** One hundred examples of debris landing positions varying with the lateral integral scale (color bar, unit: m) at wind attack angles of (**a**) 0°, (**b**) 15°, (**c**) 30°, (**d**) 45°, and (**e**) 60°.

Figure 11 shows the dimensionless impact kinetic energy at the positions where the debris impacts the ground. In general, the dimensionless impact kinetic energy increases with the longitudinal displacement of the debris. The potentially dangerous cases for dimensionless kinetic energy are those larger than 0.8, occupying a considerable proportion of the 100 cases. Furthermore, these results indicate that the kinetic energy does not have a strong relationship with the debris displacement.

Table 1 presents the statistics for the debris landing positions and dimensionless impact kinetic energy. It can be seen that the mean longitudinal displacement of different wind attack angles has a discrepancy of 2 m, first decreasing from 25.73 m and then increasing from 24.16 m with the wind attack angle. The mean value of the debris lateral displacement increases with the wind attack angle. The mean value of the dimensionless impact kinetic energy has a maximum and minimum of 0.86 and 0.76 at wind attack angles of 0° and 45°, respectively. In general, the dimensionless kinetic energy decreases with wind attack angle, except for at 60°, where it has a slight increase of 0.01 compared with 45°. The ratios of the debris longitudinal landing position and dimensionless kinetic energy at wind attack angles of 15°, 30°, 45° and 60° relative to 0° imply that the effect of the wind attack angle can be assumed by multiplying a directional factor of around 1 by a longitudinal landing position of the debris at a wind attack angle of 0°.

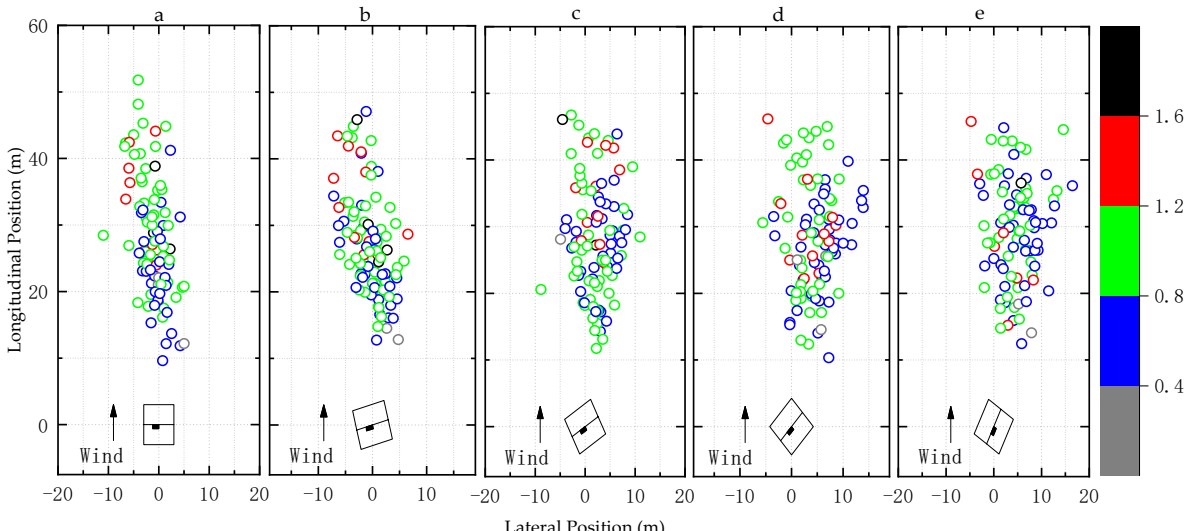

**Figure 11.** One hundred examples of debris landing position varying with dimensionless impact kinetic energy (color bar) at wind attack angles of (**a**) 0°, (**b**) 15°, (**c**) 30°, (**d**) 45°, and (**e**) 60°.

**Table 1.** Statistics of debris landing positions and dimensionless impact kinetic energy.

| Wind Attack Angle | Mean | | | Std | | | $X/X_0$ | $\overline{E}/\overline{E}_0$ |
|---|---|---|---|---|---|---|---|---|
| | $X$ | $Z$ | $\overline{E}$ | $X$ | $Z$ | $\overline{E}$ | | |
| 0° | 25.73 | −2.15 | 0.86 | 10.62 | 2.24 | 0.26 | 1.00 | 1.00 |
| 15° | 24.23 | −1.61 | 0.84 | 8.39 | 2.79 | 0.29 | 0.94 | 0.98 |
| 30° | 24.16 | 0.69 | 0.81 | 8.27 | 2.95 | 0.26 | 0.94 | 0.94 |
| 45° | 26.04 | 2.93 | 0.76 | 9.62 | 3.61 | 0.21 | 1.01 | 0.88 |
| 60° | 26.17 | 3.44 | 0.77 | 10.29 | 3.94 | 0.24 | 1.02 | 0.90 |

Figure 12 shows the probability density function and cumulative density function of the dimensionless kinetic energy of debris at five wind attack angles. In addition to the horizontal mean wind speed, the fitting results of functions at a wind attack angle of 0° indicate that the dimensionless kinetic energy corresponds well with the Generalized Extreme Value Distribution. The cumulative density function shows that about 20% of the debris dimensionless kinetic energy exceeds 1; these cases are the most dangerous if the debris impacts a building.

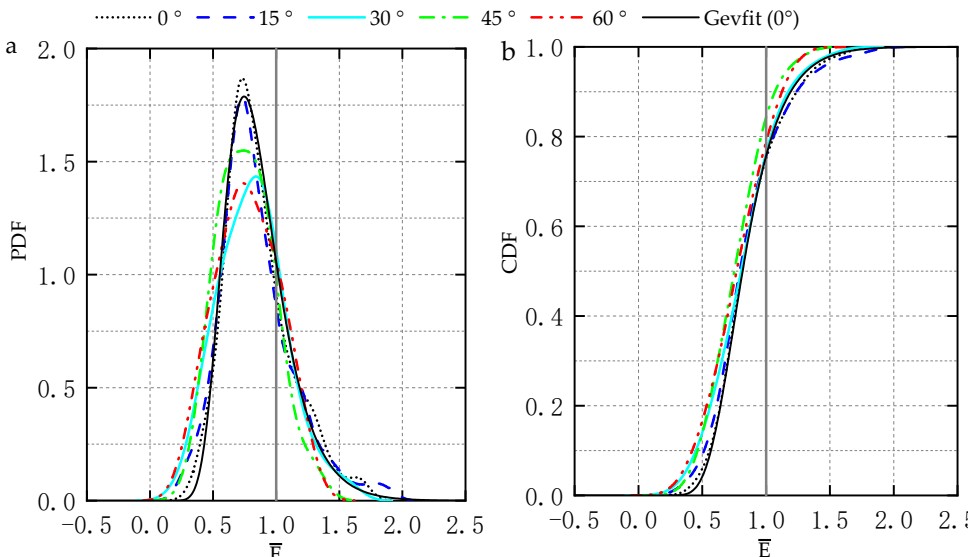

**Figure 12.** (**a**) Probability density function and (**b**) cumulative density function of the dimensionless impact kinetic energy of debris at five wind attack angles.

### 4.2. Angular Displacement and Angular Velocity

Despite the debris impact position and impact velocity, the angular displacement and angular velocity of debris flight indicate debris fight characteristics, especially for plate-type debris. As the research on the trajectory of plate-type debris is mostly based on the two-dimensional model, acknowledgment of the flight characteristics of plate-type debris in three dimensions needs to be improved.

The debris flight model presented in this paper is in three dimensions and offers convenient conditions for investigating the angular displacement and angular velocity of the debris flight trajectory. In this research, the debris rotation angular displacement and angular velocity are investigated under five different wind attack angles. Figures 13 and 14 display the effect of the wind attack angle on the debris angular displacement and velocity with the same wind speed history. The results show that the debris rotation angles, which are expressed by the Euler angles $\varphi$, $\theta$, and $\psi$, increase with the wind attack angle. The debris rotation angle is very close at the initial stage of the flight, then varies considerably with the increase in flight time. At the same time, the debris angular velocities $\omega_X$ and $\omega_Y$ increase with the wind attack angle, while the angular velocity $\omega_Z$ decreases with the wind attack angle. The debris angular velocity increases rapidly with the flight time at the beginning of the flight, and the differences in terms of maximum angular velocity are large. After that, the debris angular velocity slowly decreases to a constant value and the differences in angular velocity at different wind attack angles decrease.

As shown in Figures 13 and 14, the debris rotation angles and angular velocities are almost within the variation of 0° to 45° wind attack angle. Figures 15 and 16 present a comparison between six examples (the colored line) and the means of the 100 simulations (the solid black line) for the debris rotation angles and angular velocities at wind attack angles of 0° and 45°.

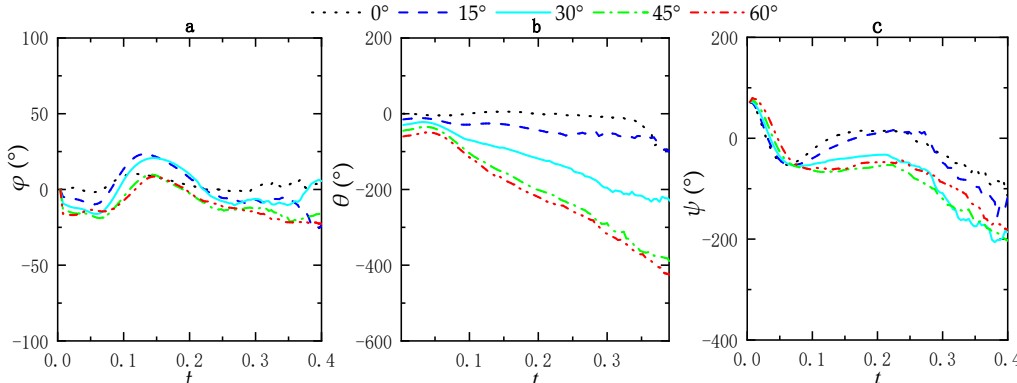

**Figure 13.** Effect of wind attack angle on debris rotation angles expressed by Euler angles: (**a**) $\varphi$, (**b**) $\theta$, and (**c**) $\psi$.

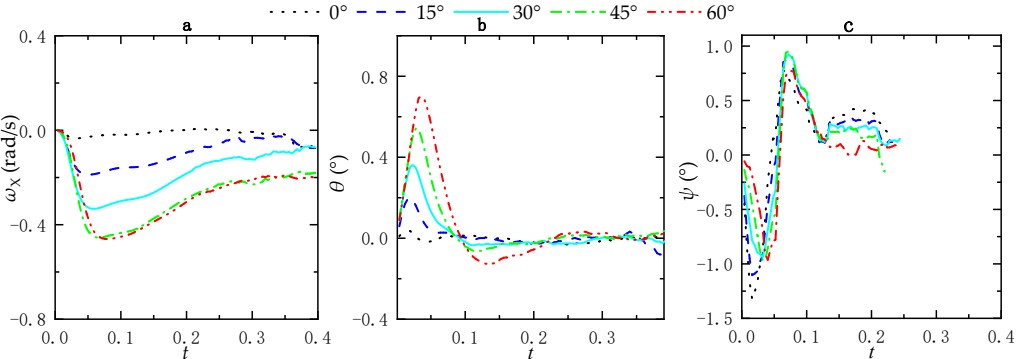

**Figure 14.** Effect of wind attack angle on debris rotation angular velocity: (**a**) $\omega_X$, (**b**) $\omega_Y$, and (**c**) $\omega_Z$.

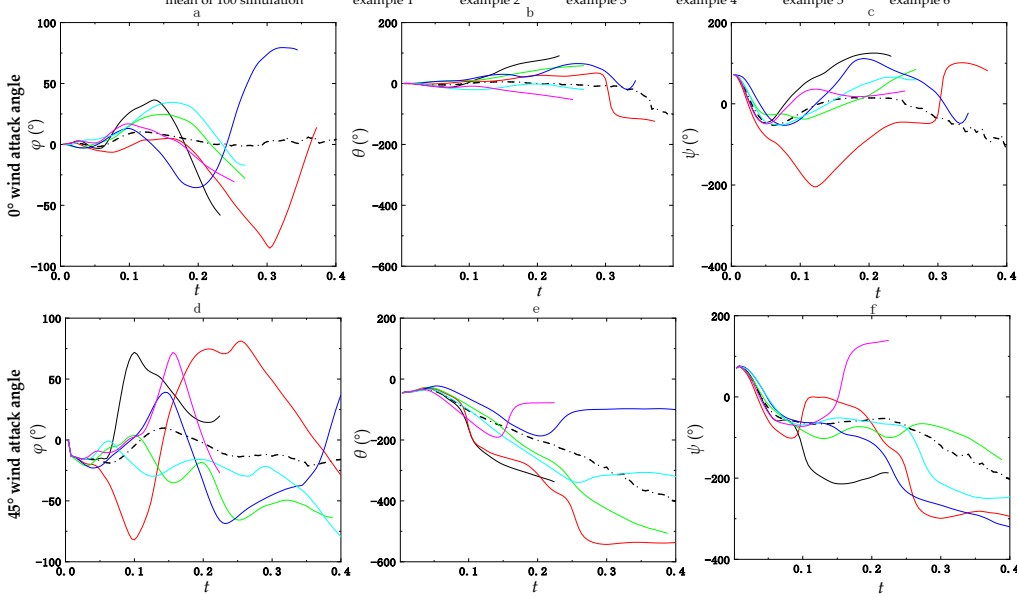

**Figure 15.** Debris rotation angles expressed by the Euler angles $\varphi$, $\theta$, and $\psi$: (**a–c**) 0° and (**d–f**) 45° wind attack angle (the colored lines stand for the single simulation examples).

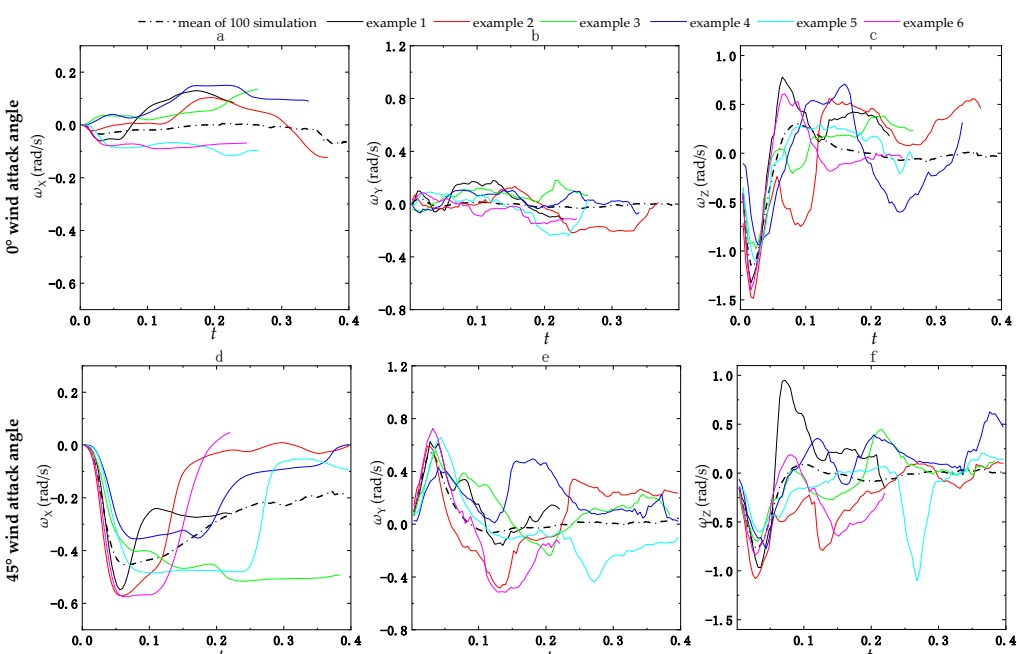

**Figure 16.** Debris rotation angular velocity expressed by $\omega_X$, $\omega_Y$, and $\omega_Z$: (**a–c**) $0°$ and (**d–f**) $45°$ wind attack angles (the colored lines stand for the single simulation examples).

Figure 15 shows that the debris angular displacement for $\varphi$ changes a little in the initial stage of the flight, after which it increases rapidly to its maximum and then increases again in the opposite direction. The debris angular displacement for $\theta$ increases steadily with flight time, then remains almost constant at its maximum. The debris rotates with the $Y$ axis in both the clockwise and anti-clockwise directions at a wind attack angle of $0°$; however, it only rotates in an anti-clockwise direction at a wind attack angle of $45°$. The debris angular displacement for $\psi$ first increases, then decreases, and at last keeps a constant value as the debris flight time increases.

In general, the angular displacements $\varphi$ and $\theta$ are small, and the changing tendency with flight time is simple in the wind direction of $0°$; however, it becomes more complex in wind directions of $45°$. The mean of 100 simulations of the debris angular displacement for $\varphi$ increases with the wind attack angle, and first increases and then decreases with the flight time. By contrast, the mean of 100 simulations of the debris angular displacement for $\theta$ increases with both the flight time and wind attack angle. The mean of 100 simulations of the debris angular displacement for $\psi$ increases with the wind attack angle, and the wind attack angle has a slightly smaller effect on the debris rotation angle for $\psi$ compared to the rotation angles for $\varphi$ and $\theta$.

Figure 16a shows that the debris angular velocity $\omega_X$ is small and increases slowly with the debris flight time at a wind attack angle of $0°$. The angular velocity $\omega_X$ increases rapidly to its maximum in the initial stage of flight, then decreases slowly and remains constant in the final stage of flight at a wind attack angle of $45°$, as shown in Figure 16d. The mean of 100 simulations of the debris angular velocity $\omega_X$ increases with the wind attack angle. The maximum and mean $\omega_X$ are about 0.12 rad/s and 0.02 rad/s, respectively, at a wind attack angle of $0°$, while the maximum and mean $\omega_X$ are about 0.6 rad/s and 0.45 rad/s, respectively, at a wind attack angle of $45°$.

Figure 16b shows that the debris angular velocity $\omega_Y$ is close to 0 and has a small fluctuation with the debris flight time at a wind attack angle of $0°$. As shown in Figure 16e, the angular velocity $\omega_Y$ increases with the wind attack angle, and the maximum value of $\omega_Y$ can reach 0.6 rad/s at a wind attack angle of $45°$. Moreover, the angular velocity $\omega_Y$ increases rapidly with the debris flight time in the initial stage of flight, then decreases in

the opposite rotation direction and varies considerably with the wind attack angle and debris flight time at a wind attack angle of 45°.

Figure 16c,f shows that the debris angular velocity $\omega_Z$ increases rapidly to its maximum in the initial stage of the flight, then quickly decreases to its opposite maximum and continues to vary slightly with the flight time at both 0° and 45° wind attack angles. The angular velocity $\omega_Z$ decreases with the wind attack angle; the maximum value of $\omega_Z$ reaches 1.5 rad/s, and its mean value from all 100 simulations is about 1.2 rad/s at a wind attack angle of 0°. The mean $\omega_Z$ first increases to its maximum, and then decreases, and at last remains constant at about 0.

## 5. Conclusions

In this paper, a turbulent wind field measured in a wind tunnel is used to consider the effects of the turbulent flow field on the trajectory of plate-type wind-borne debris. The rationality of the turbulent flow field for the simulation of debris trajectory is validated by comparing the simulated results with those from the wind tunnel testing and analytical results. Moreover, the probabilistic characteristics of the debris flight trajectory, flight velocity, rotation angular displacement, and angular velocity are investigated under five different wind attack angles. The main conclusions of this work are as follows:

- The wind attack angle has a significant effect on the flight velocities and trajectories of the debris. At a wind attack angle of 0°, the debris lands within a relatively narrow lateral displacement. Moreover, the debris impact velocity increases with longitudinal displacement, and many pieces of debris impact the ground with a dimensionless velocity larger than 1. The landing positions of the debris are more concentrated in small wind attack angles.
- For wind attack angles between 0° and 60°, the mean value of the dimensionless impact kinetic energy has its maximum and minimum at 0.86 and 0.76 at 0° and 45°, respectively. The cumulative density function shows that about 20% of the debris dimensionless kinetic energy exceeds 1; these cases are the most dangerous for debris impacting a building.
- The debris rotation angle $\varphi$ is more influenced by the uncertainty of the debris trajectory than the wind attack angle. The debris rotation angular velocities $\omega_X$ and $\omega_Y$ increase with the wind attack angle, and the mean of $\omega_X$ and $\omega_Y$ is very close to 0 under a wind attack angle of 0°. On the other hand, the debris rotation angular velocity $\omega_Z$ decreases with the wind attack angle; the maximum of the mean angular velocity $\omega_Z$ is 1.2 rad/s under a wind attack angle of 0°.

**Author Contributions:** Methodology and writing—original draft preparation, F.W. and P.H.; resources, R.Z.; data curation, Z.Z.; writing—review and editing, H.W. and M.S.; supervision, P.H. and Y.X. All authors have read and agreed to the published version of the manuscript.

**Funding:** This research was funded by Shanghai Sailing Program grant number 21YF1439400, Scientific Research Project Plan of Shanghai Municipal Commission of Housing and Urban–Rural Development grant number Hujianke2023-002-029 and the APC was funded by Program of Shanghai Academic/Technology Research Leader grant number 22XD1433300.

**Data Availability Statement:** Derived data supporting the findings of this study are available from the corresponding author on request.

**Acknowledgments:** This project was sponsored by the Shanghai Sailing Program (21YF1439400), Program of Shanghai Academic/Technology Research Leader (22XD1433300), and Scientific Research Project Plan of Shanghai Municipal Commission of Housing and Urban–Rural Development (Hujianke2023-002-029), which are gratefully acknowledged.

**Conflicts of Interest:** The authors declare no conflict of interest.

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
