# Peer review of "Predicting Trajectories of Plate-Type Wind-Borne Debris in Turbulent Wind Flow with Uncertainties"

_infrastructures, doi:10.3390/infrastructures8120180_

Round 1
Reviewer 1 Report
Comments and Suggestions for Authors
This paper discusses a very interesting topic with a good scientific approach based on both, experimental and numerical results. The topic is interesting for readers.
I suggest a list of symbols. Some variables are not defined in the manuscript, as for example Eq(1).
Fig. 3 should be improved, maybe 0° should be added on the plan view.
In Fig.12 the PDF and CFD could be fitted by numerical PDF (Guassian, Lognormal,...etc). Maybe the CDF is enough to discuss results.
Figs 13-15 quality should be improved.
A crucial point should be dicsussed carefully: the experimental error propagation. Authors can be interested to read the paper below and to discuss the approach.
F. Rizzo, F. Ricciardelli, G. Maddaloni, Bonati A., A. Occhiuzzi, 2020. Experimental error analysis of dynamic properties for a reduced-scale high-rise building model and implications on full-scale behavior. Journal of Building Engineering, 28.
The wind speed histoy can be divided in subintervals and analyses can be repeated many times to investigate the variability of magnitdes. Please give some comments
Reviewer 2 Report
Comments and Suggestions for Authors
The manuscript investigated the 3D trajectories of plate-type wind borne debris in turbulent wind field via numerical simulation method. In the process of numerical simulation, a 3D probabilistic trajectory model was developed, and the turbulent wind field was measured in the wind tunnel. After the validation of the probabilistic trajectory model, the probability characteristics of debris impact position, impact velocity and kinetic energy, debris angular displacement and angular velicity were analyzed in detail. Specific comments below need to be addressed before publication of this manuscript.
1. (i= x, y, z) should be added in the formula (10).
2. How the standard deviation wind speeds σi were calculated? Please presnet the formulas.
3. As the expected wind speed spectra to validate the measured wind speed spectra of the three components, it is advised to present the formulas for the von Karman spectra in three directions.
4. What are the 9 cases in Figure 7? You need show the legend for the curves.
5. The reference [26] is missing.
Comments on the Quality of English LanguageThere are a few wrong spelling and expressions in the manuscript. It is advised to ask a native English speaker to impove English, and a thorough and careful reading may be done to remove English issues.
Round 2
Reviewer 1 Report
Comments and Suggestions for Authors
Authors have revised their paper correctly.